# Improving the Bond Strength of Radiographically Tagged Caries Lesions In Vitro [note 1]

**DOI:** 10.3390/ma13173702

**Published:** 2020-08-21

**Authors:** Sophia Toelle, Agnes Holtkamp, Uwe Blunck, Sebastian Paris, Falk Schwendicke

**Affiliations:** 1Department of Oral Diagnostics, Digital Health and Health Services Research, Charité—Universitätsmedizin Berlin, Aßmannshauser Str. 4-6, 14197 Berlin, Germany; sophia.toelle@charite.de; 2Department of Operative and Preventive Dentistry, Charité—Universitätsmedizin Berlin, Germany, Aßmannshauser Str. 4-6, 14197 Berlin, Germany; agnes.holtkamp@charite.de (A.H.); uwe.blunck@charite.de (U.B.); sebastian.paris@charite.de (S.P.)

**Keywords:** bonding, caries, dental, diagnostics, radiography, residual caries

## Abstract

After selective carious tissue removal, residual carious lesions remain radiographically detectable. Radiopaque tagging resolves the resulting diagnostic uncertainty but impedes bond strengths of adhesives to tagged dentin. We developed a protocol mitigating these detrimental effects. A 30%/50%/70% SnCl_2_ solution was dissolved in distilled water or a 30%/50%/90% ethanol solution (E30/60/90) and applied to artificially induced dentin lesions. Tagging effects were radiographically evaluated using transversal wavelength-independent microradiography (*n* = 6/group). Groups with sufficient tagging effects at the lowest SnCl_2_ concentrations were used to evaluate how tagging affected the microtensile bond strength of a universal adhesive (Scotchbond Universal) to sound and carious dentin (*n* = 10/group). Two different protocols for removing tagging material were tested: 15 s phosphoric acid etching and 5 s rotating brush application. Scanning/backscattered electron microscopy (SEM/BSE) and energy-dispersive X-ray spectroscopy (EDS) were used to assess surfaces after tagging and removal. The most promising removal protocol was revalidated microradiographically. Tagging significantly increased the radiopacity, with consistent effects for 30% SnCl_2_ dissolved in water or E30. Microscopically, tagged surfaces showed a thick carpet of SnCl_2_, and tagging reduced bond strengths significantly on carious dentin but not on sound dentin (*p* < 0.01). On carious dentin, removal of tagging material using acid etching and rotating brush was microscopically confirmed. Acid etching also mitigated any bond strength reduction (median: 21.3 MPa; interquartile range: 10.8 MPa) compared with nontagged dentin (median: 17.4 MPa; interquartile range: 20.6 MPa). This was not the case for brushing (median: 13.2 MPa; interquartile range: 13.9 MPa). Acid etching minimally reduced the radiographic tagging effect (*p* = 0.055). Phosphoric acid etching reduces the detrimental bond-strength effects of tagging without significantly decreasing radiographic tagging effects when using a universal adhesive.

## 1. Introduction

For deeper carious lesions, selective carious tissue removal is recommended, which aims to avoid pulp exposure and pulpal complications and thus preserve pulp vitality and health [1,2]. In selective removal, carious tissue is intentionally left close to the pulp during the excavation process, while in the peripheral areas of the cavity, excavation is performed until firm or hard dentin remains, ensuring restoration longevity and a tight seal of the cavity [2]. Sealed carious lesions are inactivated by bacteria being deprived of dietary carbohydrates after depleting intracellular polysaccharide depots [3,4]. There is a growing body of evidence supporting selective removal [5], while admittedly the majority of studies have been conducted in the primary and not the permanent dentition [6,7].

One drawback of this strategy, though, is sealed dentin remaining detectable as radiolucency beneath the restoration [8]. The resulting diagnostic uncertainty might be why the majority of dentists worldwide have not accepted selective removal as standard yet [9]. Radiopaque tagging of carious dentin prior to the restoration placement has been suggested to overcome this uncertainty: tagged carious dentin was no longer detectable on radiographs, while tagging effects were shown to decrease in case the described inactivation of the lesion was not achieved [8].

For tagging, radiopaque substances like stannous chloride or fluoride have been suggested and employed in higher concentrations. These also seem to have antibacterial effects [10]. A key and notable limitation of this concept at present is that such high-concentration tagging was found to significantly decrease the bond strengths of dental adhesives to both carious and sound dentin (lowly concentrated stannous chloride has been found to not negatively affect bond strengths) [11]. Similar findings were reported for treatment of carious dentin using other metal ions, like silver diamine fluoride (SDF) [12]; for SDF, rinsing off or refreshing the surface mitigated these detrimental effects [13]. Resolving the issue of decreased bond strengths may allow implementing radiopaque tagging as a standard operative step after selective carious tissue removal and before restoring the cavity [10].

This study aimed to evaluate different strategies for improving bond strengths of a universal adhesive after tagging carious dentin. These strategies included optimization of the concentration and solvent of the tagging material to balance tagging against bond-strength effects, along with an application of different steps for cleaning the tagged dentin surface. We hypothesized that eventually, by using the developed protocol, there would be no significant difference in bond strength of a universal adhesive to both tagged and nontagged dentin surfaces.

## 2. Methods

### 2.1. Study Design

SnCl_2_ was dissolved in different concentrations (30%, 50%, 70%) in different media, namely distilled water (Aqua destillata, AQ) and 30%, 60%, and 90% ethanol (E30/60/90), and applied to tag artificially induced dentin caries lesions. The rationale was the different solubility of metal ions in different solvents and solvents of different concentrations, as demonstrated before [8].

Tagging effects were radiographically evaluated using transversal wavelength-independent microradiography (T-WIM). Groups with sufficient tagging effects at the lowest SnCl_2_ concentrations were used to evaluate how tagging affected the bond strength of adhesive restorations to sound and carious dentin when submitted to microtensile bond strength (μ-TBS) measurements. Two different protocols for removing bond-strength-detrimental tagging material from the dentin surface were tested: phosphoric acid etching and rotating brushing. Scanning and backscattered electron microscopy (SEM, BSE) and energy-dispersive X-ray spectroscopy (EDS) were used to assess surfaces after different tagging and removal protocols. The tagging effects of the most promising removal protocol were eventually validated microradiographically once more, using T-WIM. The sample size for the different steps was not decided based on statistical assumptions; it was based on experience from the discussed prior studies in the field.

### 2.2. Tagging of Caries Lesions

Two hundred sound extracted human permanent molars were used for the study. Molars were obtained under an ethics-approved protocol (ethics committee of the Charité EA4/102/14) and informed consent. Four hundred dentin samples (3 × 2 × 1 mm) were cut from the molars (Band Saw EXAKT 300 CL, EXAKT Technologies, Norderstedt, Germany), exposing coronal dentin, which was subsequently polished up to 4000 grit (SiC paper, Struers, Willich, Germany). The cutting was stopped and polishing started when the coronal surface was free of any enamel remnants to ensure a sufficient dentin layer (> 1 mm thickness) was left between the polished coronal surface and the pulpal area. It should be noted that, clinically, tagging would not necessarily be performed on only outer or middle coronal dentin; rather, it would be performed on the deepest dentin areas. This was not experimentally feasible, though.

Afterwards, samples were separated axially and prepared for T-WIM. To induce locally defined artificial carious lesions, half of the coronal surface was covered using nail varnish (Riva-De-Loop, Rossmann, Burgwedel, Germany). Lesions were then chemically generated in unprotected areas by storing samples in demineralizing solution at pH 5.3 and 37 °C under agitation using a validated protocol [14] for 9 weeks. The integrated mineral loss (ΔZ) at baseline was assessed using T-WIM (see below). Demineralized areas were air-dried for 5 s, and SnCl_2_ (Sigma Aldrich, St. Louis, MI, USA) solutions were applied twice for 15 s with a microbrush. Any excess solution was removed using microbrushes. SnCl_2_ was dissolved in different media (distilled water and ethanol) (J.T.Baker, Deventer, The Netherlands) in different concentrations (30%, 50%, 70%).

### 2.3. Transversal Wavelength-independent Microradiography

The radiographic effect of tagging (ΔΔZ) was assessed using T-WIM [15] as previously described [8]. For T-WIM, 35 mm films (B/W positive, Fujifilm, Tokyo, Japan) were used, and microradiographs were assessed microscopically (Axioplan 60318, Zeiss, Oberkochen, Germany) and digitalized (CFW1312M, Scion, Frederick, MD, USA). ΔΔZ was measured using TMR 2000 (2.0.27.2, UMCG, Groningen, The Netherlands) and calculated as follows: ΔΔ Z (%) = −(ΔZ_tagging_ − ΔZ_baseline_) × ΔZ^−1^. Positive values indicate a reduced radiolucency.

### 2.4. Bond Strength

To test bond strength, 40 of the extracted molars were first embedded in methacrylate resin (Technovit 4071, Heraeus Kulzer, Hanau, Germany). The occlusal surfaces of the molars were exposed and polished as described above. On the polished surface of 25 teeth, artificial lesions were again chemically induced [14] using acetic acid solution (pH 5.3, 37 °C) as described. Storage of the remaining samples was conducted in distilled water. Radiopaque materials were applied to a total of 22 carious teeth and 12 sound teeth. The rationale for using sound dentin as control was that tagging is likely to also affect non-carious areas, too, and it is relevant how such accidental tagging of sound surfaces affects bond strengths. Application was performed twice (2 × 20 μL) using a microbrush for a total of 30 s, and then blotted dry using another microbrush. Three sound and three carious teeth were used as nontagged controls.

A universal adhesive system (Scotchbond Universal, 3M, St. Paul, MN, USA) was rubbed on all teeth for 20 s moving using an application tip, air-dried gently, and light-cured for 15 s. A resin composite (Ceram.X Spectra, Dentsply, Konstanz, Germany) was then placed according to the manufacturer’s instructions. Composite was placed in increments until a total thickness of 6 mm was achieved. Light-curing was performed with an LED curing light (Valo, Ultradent, Salt Lake City, UT, USA) at an intensity of 1400 mW/cm^2^ for 40 s from a distance of <1 mm.

After 24 h storage in distilled water at 37 °C, samples were sectioned using the Isomet 1000 (Buehler, Lake Bluff, IL, USA) at low speed and under water-cooling, resulting in sticks with a rectangular cross-section (ca. 1 mm^2^). From each tooth, three sticks were randomly selected; the sample size per group was *n* = 10. For microtensile testing, sticks were attached to the test assembly using Tetric EvoFlow (Ivoclar Vivadent, Schaan, Liechtenstein). No pretesting failures occurred. The tensile load was applied at a crosshead speed of 0.5 mm/min (Zwick, Ulm, Germany). The maximum force recorded before any fracture occurred was used to estimate the microtensile bond strength. Fragments were finally assessed under a stereomicroscope (Stemi, Zeiss) at 45× magnification in order to discriminate between different failure modes: (1) cohesive failure in dentin, (2) adhesive failure between adhesive system and dentin, (3) adhesive failure between adhesive system and composite, (4) mixed adhesive failure (failure modes 2 and 3), and (5) cohesive failure in composite.

### 2.5. Scanning and Back-scattered Electron Microscopy (SEM, BSE) and Energy Dispersive X-ray Spectroscopy (EDS)

Samples were air-dried for SEM and BSE or carbon-coated for EDS. SEM and EDS were performed using a CamScan MaXim microscope (CamScan, Cambridge, UK). A Bruker XFlash 6 30 detector was used, and the analyses were performed using the ESPRIT 2.0 software (Bruker Nano, Berlin, Germany). The acceleration voltage was 15 kV. BSE was performed on Phenom SEM at an acceleration voltage of 10 kV. The acquisition time for the mapping was 300 s.

### 2.6. Statistical Evaluation

Statistical analysis was performed using SPSS 20 (IBM, Armonk, NY, USA). Normal distribution was assessed using the Shapiro–Wilk test. Relative tagging effects on mineral loss ΔZ (ΔΔZ in %) were analyzed using generalized linear modeling and two-sided Mann–Whitney U-test. Positive ΔΔZ values indicate reduced translucency, i.e., effective tagging. Level of significance was set at α = 0.05.

## 3. Results

Tagging significantly increased the radiopacity, and all groups except E60-30% and E90-30% showed significant tagging effects (Figure 1, *p* < 0.05). Overall, tagging effects were more pronounced in AQ and E30 than E60 and E90 (*p* < 0.05). There was a dose–response relationship for nearly all groups, with higher SnCl_2_ concentration resulting in higher radiopacity. However, differences were not consistently significant, which is why further experiments focused on optimizing an application protocol for AQ and E30, both with 30% SnCl_2_, were necessary.

Tagging effects were also detectable via SEM (Figure 2) and BSE and EDS (Figure 3). Tagged surfaces showed a thick carpet of dense particles which were confirmed as SnCl_2_; these particles were crystalline (e.g., Figure 2b) and were observed to be either homogenously distributed (for AQ) or occurring in large insulae (for E30).

Different removal protocols were tested, as described. Both protocols visibly reduced the microscopic presence of tagging material (Figure 2), with SnCl_2_ particles remaining either within the tubules (for acid etching, e.g., Figure 3f,g) or in striae (for brushing removal, e.g., Figure 2i,k).

Bond strengths were significantly higher on sound than on carious dentin (Figure 4; *p* < 0.05 (Mann–Whitney)). Tagging reduced bond strengths significantly on carious dentin, but not on sound dentin, with significantly lower bond strength on sound dentin observed when E30 was used for tagging when compared with AQ (*p* < 0.05). Using phosphoric acid after tagging increased bond strengths, with acid etching leading to higher bond strengths on sound dentin compared with an untreated control, even on tagged surfaces (*p* < 0.05). On carious dentin, acid etching led to bond strength of AQ-tagged samples being not significantly different from those in untreated controls. Brushing did not lead to a significant bond strength increase in sound dentin, and it also did not mitigate the effects of tagging on carious dentin (*p* < 0.01). For all samples, adhesive failures between the adhesive system and dentin occurred (i.e., while there were variations in bond strengths, the failure mode did not differ between groups).

When acid-etched samples were assessed for their radiopacity (Figure 5), a minimal decrease in tagging effect was notable; the median (IQR) ΔΔZ compared with untreated controls was 73.2% (29.1%) in tagged and 51.6% (18.8%) in tagged and etched lesions (*p* = 0.055).

## 4. Discussion

Selective carious tissue removal was found beneficial for maintaining pulp vitality in teeth with deeper carious lesions. The resulting radiolucency beneath the restoration, however, poses diagnostic uncertainties for dentists. Radiographic tagging was shown to positively mitigate this problem but detrimentally affect bond strengths to the tagged dentin. Within the present study, we demonstrated that a concentration of 30% of SnCl_2_, dissolved in water or 30% ethanol, showed a reliable and sufficient tagging effect while limiting the reported impact on bond strength of the used universal adhesive. Nevertheless, bond strengths decreased by 30–60%, i.e., significantly and possibly clinically disadvantageously, when using a universal adhesive. However, subsequent phosphoric acid etching was able to resolve this issue, without significant decreases in tagging effects. We therefore accept our hypothesis.

This study has a number of strengths and limitations. First, in line with previous work in this field [8,10], we relied on established in vitro test methodologies and triangulated our findings from one test method (e.g., µ-TBS) with those from others (e.g., SEM, EDS). Further assessment methods like interfacial SEM or TEM may be used to elucidate the described effects in more detail. Second, we employed easy-to-apply protocols to optimize bond strengths; translating these protocols into clinical application will likely be unproblematic. Third, as a limitation, we worked with sound dentin and artificial carious lesions in a modeling system. However, both the lesion induction protocol as well as the model were validated in previous studies. We also used only one specific universal adhesive system, while many more are available. Our findings might not fully translate to all universal adhesive systems or to other (non-universal) adhesives, as discussed below. Fourth, we only used one specific tagging material, SnCl_2_, while others might be available. This material, however, was used in previous studies and found to be beneficial in a range of aspects (see below) [8,10]. Last, we tested tagging effects using T-WIM, and not in clinical radiographic applications. However, we have demonstrated translucency changes detected in T-WIM to correlate with those detected by conventional radiography [8] and to match tagging effects of natural lesions (and not only artificial lesions, like in the present study). Notably, it is unclear if the differences found in radiolucency between etched and non-etched tagging effects will be clinically relevant or not (even if they were not significant in our study). Furthermore, the present study did not assess the safety of the application, while in another evaluation (unpublished), we evaluated the pulpal pH changes induced by tagging, for instance, in an in vitro pulp model, without finding significant impacts. Further related issues like possible cell toxicity should be evaluated.

A range of aspects need to be discussed. In our evaluation, AQ showed tagging effects similar to those of E30 without relevant differences between SnCl_2_ concentrations. In contrast, dissolving SnCl_2_ in 60% or 90% ethanol resulted in less reproducible tagging effects, and lower tagging when employing 30% SnCl_2_, than in AQ and E30. This is likely due to the inferior solubility of SnCl_2_ in high ethanol concentrations, the result being an inhomogeneous, clustered precipitation of SnCl_2_ on dentin and hence, reduced tagging effects. These presumed segregation effects were confirmed when dentin surfaces were evaluated using SEM and BSE. In all groups, we observed SnCl_2_ crystal precipitation. However, in ethanol, these precipitations were less homogenously distributed, mainly in islands or striae of material.

These precipitations are the reason for the impeded bond strengths of dental adhesives. On carious dentin, we observed mean bond strength reductions of 30–60% when compared with nontagged surfaces. Tagging did not induce such reductions on sound dentin, indicating that significant precipitation may not occur there. The applied removal strategies had notable effects on the bond strengths. On sound nontagged dentin, phosphoric acid etching increased bond strengths, likely by removing the smear layer (resulting from sample polishing) and opening up the collagen network, thereby improving the formation of a hybrid layer as well as the interaction between the universal adhesive’s 10-MDP and the exposed dentin. This positive effect of etching was not noted in carious nontagged dentin, where the carious attack has likely removed the smear layer. On sound tagged surfaces, we found acid etching to again improve bond strengths to “super-normal” levels and to reinstate the bond strengths to those found in carious untagged dentin. This is relevant, as in a previous study, we did not find etching to have such effect when applying a conventional etch-and-rinse adhesive (in this case, OptiBond FL) afterwards [10]. We ascribe this differential effect to the presence of 10-MDP in the universal adhesive, with the positive effects of 10-MDP only being effective when removing the precipitates and allowing chemical interaction with the dentin surface. Using a rotating brush did not have such beneficial effects on sound or carious, tagged or untagged surfaces. We attribute this to the fact that the brush does not remove the smear layer or any precipitates, but rather redistributes them in “streets” of the material, following the rotating movement (see Figure 3).

Consequently, we employed acid etching to remove precipitations and reinstate bond strengths and re-evaluated the resulting tagging effects. We found that phosphoric acid etching did not significantly decrease tagging effects. Notably, and as mentioned, the absence of statistical significance should not be taken as indicative of a lack of relevance, as it might also be grounded in the limited statistical power. Further experiments using conventional radiographic assessments are warranted.

The developed protocols are easy to apply and likely not time-consuming. Selective phosphoric acid etching of enamel is often recommended even when using universal adhesives (such as the one we used) in self-etching mode to obtain ideal bond strengths on enamel [16]. Our protocol suggests combining an etch-and-rinse approach after tagging to optimize the strength of bonds to both enamel and dentin. A range of questions, however, remain: First, the pH of the used tagging solutions has been found to be low [8], and possible detrimental effects when applied onto pulpo-proximal dentin should be explored (as discussed, no relevant pH decreases were observed in a modeling system). Second, we did not test the differential effect of tagging lesions of different depths. This, however, was done in a previous study [10], and it is also likely that some tagging effect will be exerted in deeper lesions by the used tagging solutions. Given the high concentration of the tagging material, even tagging outer carious dentin layers has been found to allow the masking of deeper lesions. Third, the longevity of tagging has not been fully explored. A previous study, however, tried to wash out tagging material from carious dentin using thermocycling and failed when the carious lesion remained arrested (i.e., the pH in dentin did not fall below a certain threshold) [8]. Clinical observations over long-term periods may be required. These studies should also evaluate if tagging (using the outlined protocol) really reduces the rate of clinicians who unnecessarily intervene on carious lesions sealed after selective removal, as demonstrated before. Last, the effect on bacteria may be tested in more detail. This is especially relevant given that sealing of lesions is thought to deprive cariogenic bacteria of nutrition, while asaccarolytic bacteria have been argued to remain viable, leading to possible pulp inflammation [17]. In that direction, the effects of tagging on pulpal cells should also be assessed, as discussed.

## 5. Conclusions

When using the universal adhesive Scotchbond Universal, phosphoric acid etching of radiopaque-tagged dentin mitigated the detrimental bond-strength effects of tagging while preserving the beneficial radiographic effects. A separate dentin etching step should be recommended after tagging carious lesions.

## Figures and Tables

**Figure 1 materials-13-03702-f001:**
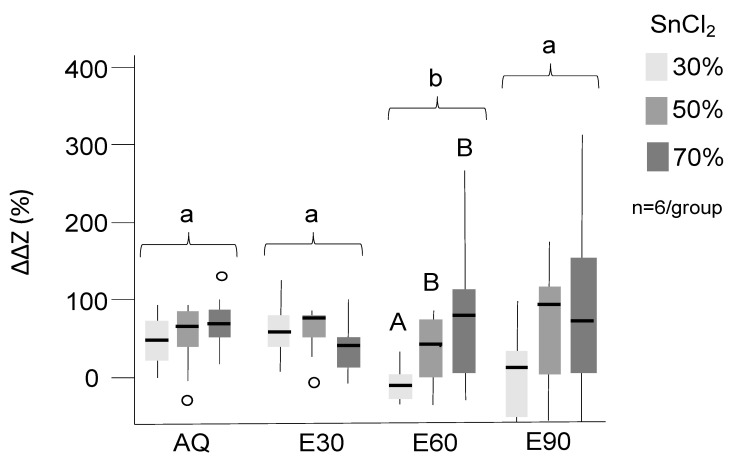
Mineral loss differences in % compared to untreated lesions. Positive values indicate reduced translucency and effective tagging. SnCl_2_ (30%, 50%, 70%, indicated by different grey shades) was dissolved in distilled water (AQ) and 30%, 60%, and 90% ethanol (E30–90). Significant differences between tagging groups (*p* < 0.05) are indicated by different letters; upper case letters are used to indicate differences within the solutions while lower case letters indicate difference between the solutions. Box and line: interquartile range and median; whiskers: minimum/maximum; circles: outliers; n: sample size.

**Figure 2 materials-13-03702-f002:**
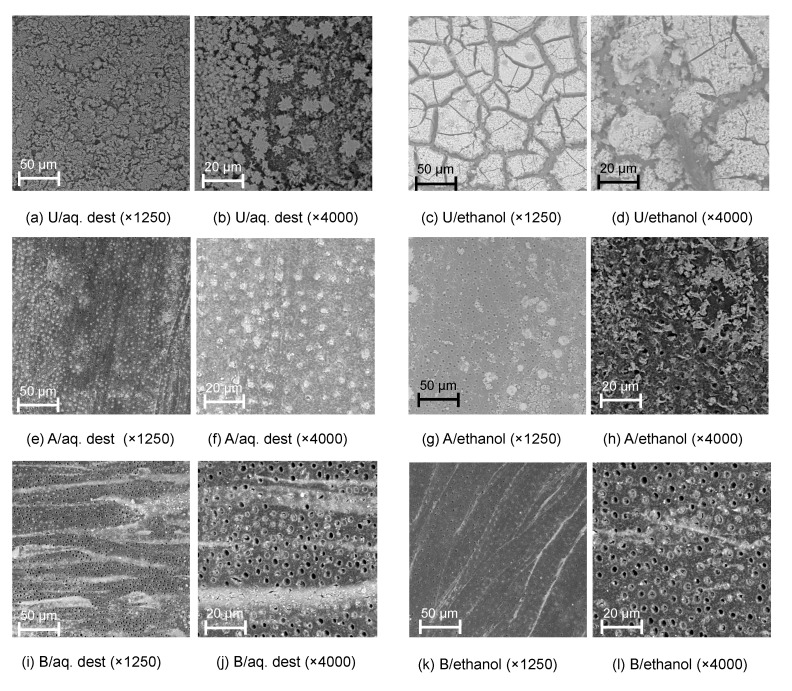
Scanning electron microscopy images of dentin tagged with 30% stannous chloride dissolved in distilled water (aq. dest.; **a**,**b**,**e**,**f****,i**,**j**) or ethanol (**c**,**d**,**g**,**h**,**k**,**l**) without further removal (untreated, U; **a**–**d**), or removal using acid (A; **e**–**h**) or rotating brushes (B; **i**–**l**), at different magnifications.

**Figure 3 materials-13-03702-f003:**
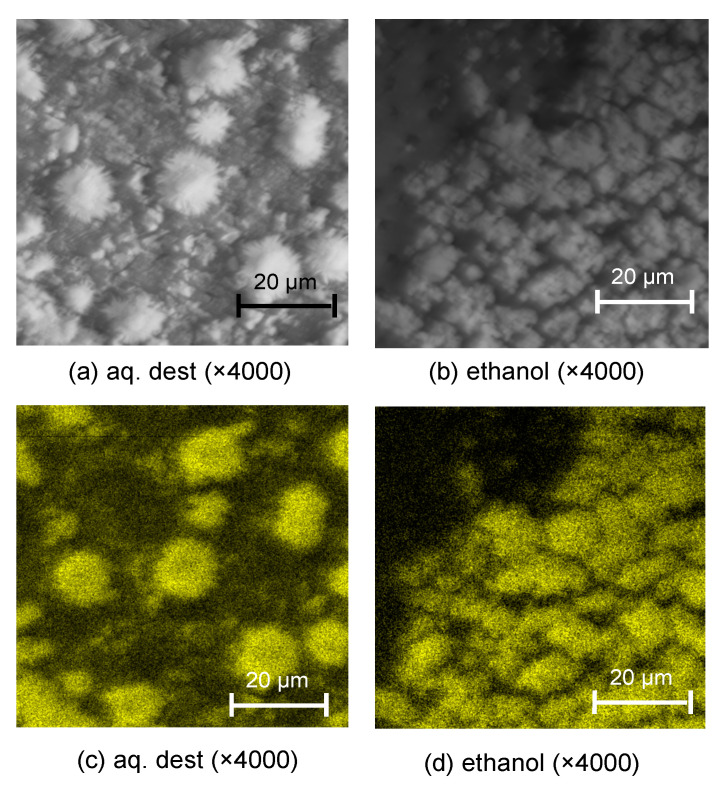
Backscattered SEM (BSE) and EDS analysis. BSE microscopy at 10 keV (**a**,**b**) and EDS (**c**,**d**) analysis of dentin tagged with stannous chloride dissolved in distilled water (aq. dest.) or ethanol. In BSE, brighter crystalline structures covering the dentin can be detected; in EDS, these plaques are confirmed as containing high concentrations of Sn (indicated by yellow color).

**Figure 4 materials-13-03702-f004:**
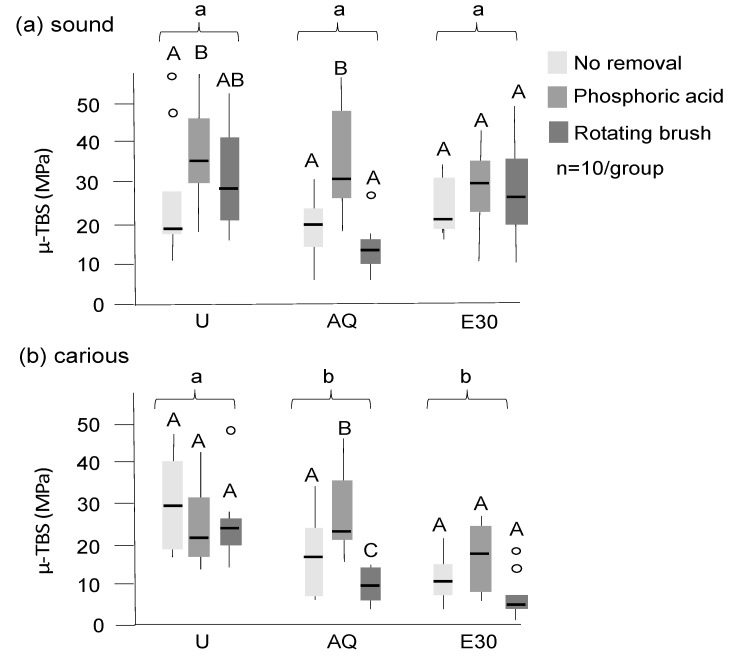
Microtensile bond strength (µ-TBS) of adhesive restorations on sound (**a**) and carious (**b**) dentin after different treatments (in MPa). Untreated sound and carious dentin (U), dentin tagged with 30% SnCl_2_ dissolved in distilled water (AQ), and 30% ethanol (E30) are shown, with tagging material not being removed or being removed with phosphoric acid or a rotating brush (different grey values). Significant differences (*p* < 0.05) between groups and within groups are indicated by different lower- and uppercase letters, respectively. Box and line: interquartile range and median; whiskers: minimum/maximum; circles: outliers; *n*: sample size.

**Figure 5 materials-13-03702-f005:**
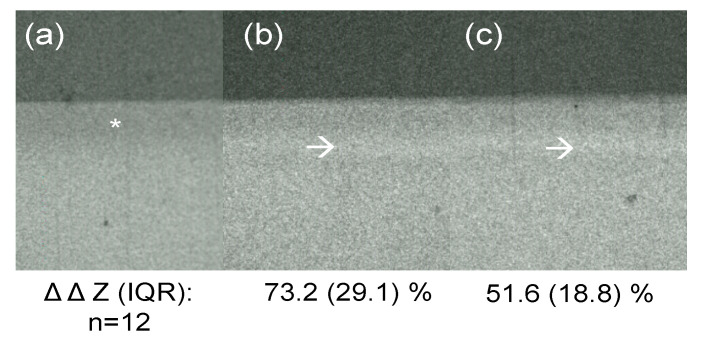
Transversal wavelength-independent microradiography (T-WIM) images of a demineralized and untreated lesion (**a**), a lesion tagged with 30% SnCl_2_ dissolved in distilled water (**b**), and the same lesion after phosphoric acid etching (**c**). Mineral loss differences (median, interquartile range (IQR), in %) compared with untreated lesions are given. n: sample size. Asterisk: induced lesion (radiolucency). Arrows: tagging material (radiopacity).

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
