# Peer review of "Improving the Bond Strength of Radiographically Tagged Caries Lesions In Vitro†"

_materials, 2020, doi:10.3390/ma13173702_

Round 1

Reviewer 1 Report

Dear authors,

I'm very interested in this paper and your coworker's previous papers.

Beacause we are facing aging societies all over the world especially in Japan.

In terms of root caries, it is difficult to remove caries or to fill the cavity with restorative materials.

So even in that sense, this work has very important meaning, I believe.

In the present paper, I have some comments.

First, you used coronal part of dentin, but you didn't explain the details. Please mention the details about which part of coronal dentin you got. I think you might have gotten two dentin blocks from one coronal part of tooth, right? How deep are they?

Second, in the results, after getting bond strength you checked failure modes and said all samples had adhesive failures. However, from your results of µTBS, there were some variations?

Last, you only show 12 references including your coworker's papers in this paper. So please reference other papers.

Author Response

Reviewer 1

Comment: First, you used coronal part of dentin, but you didn't explain the details. Please mention the details about which part of coronal dentin you got. I think you might have gotten two dentin blocks from one coronal part of tooth, right? How deep are they?

Our response: This was explained in more detail: “The cutting was commenced and polishing started just when the coronal surface was free of any enamel remnants to ensure a sufficient dentin layer (> 1mm thickness) being left between the polished coronal surface and the pulpal area. Note that clinically, tagging would not necessarily be performed only to such rather outer or middle coronal dentin, but the deepest dentin areas. This was experimentally not feasible.” 

Comment: Second, in the results, after getting bond strength you checked failure modes and said all samples had adhesive failures. However, from your results of µTBS, there were some variations?

Our response: There were variations in bond strength, but not failure mode. This was once more pointed out.

Comment: Last, you only show 12 references including your coworker's papers in this paper. So please reference other papers.

Our response: The introduction and discussion were expanded and further references introduced.

Reviewer 2 Report

Dear authors,

  I read your manuscript with interest. I want only to highlight some minor aspects to be fixed:

  1. Please avoid to self-cite about "literature on selective removal" and try to cite other studies about the topic;
  2. I do not agree with the affirmation that "bacteria are deprived from dietary carbohydrates when carious lesions are sealed" as it is demonstrated that intracellular polysaccharide is produced by Streptococcus Mutans in case of excess glucose and then used in starvation. Please correct.
  3. How was the sample size calculated? No power analysis is given...
  4. Reference 3 is not citable as it is not a published article

Regards

Author Response

Reviewer 2

Comment: Please avoid to self-cite about "literature on selective removal" and try to cite other studies about the topic.

Our response: The introduction and discussion were expanded and further references introduced.

Comment: I do not agree with the affirmation that "bacteria are deprived from dietary carbohydrates when carious lesions are sealed" as it is demonstrated that intracellular polysaccharide is produced by Streptococcus Mutans in case of excess glucose and then used in starvation. Please correct.

Our response: This was changed accordingly.

Comment: How was the sample size calculated? No power analysis is given...

Our response: This was explained: “The sample size for the different steps was not decided based on statistical assumptions, but experience from the discussed prior studies in the field.”

Comment: Reference 3 is not citable as it is not a published article

Our response: This was a mistake, the article is citable and full details were added.